# Carbon-Based Synthesized Materials for CO₂ Adsorption and Conversion: Its Potential for Carbon Recycling

**Tuan-Dung Hoang** [1,2], **Suhaib A. Bandh** [3], **Fayaz A. Malla** [4,*], **Irteza Qayoom** [5], **Shahnaz Bashir** [6], **Suhail Bashir Peer** [7] **and Anthony Halog** [8,*]

1 School of Chemistry and Life Science, Hanoi University of Science and Technology, No. 1 Dai Co Viet, Hai Ba Trung, Hanoi 100000, Vietnam; tuandunghoang@gmail.com or tuandung@vnu.edu.vn
2 Vietnam National University, Hanoi, VNU Town, Hoa Lac, Thach That District, Hanoi 155500, Vietnam
3 Department of Higher Education, Government of Jammu and Kashmir, Srinagar 190001, Jammu and Kashmir, India
4 Department of Environmental Science, Government Degree College Tral, Pulwama 192123, Jammu and Kashmir, India
5 Centre for Interdisciplinary Research & Innovations, University of Kashmir, Srinagar 190006, Jammu and Kashmir, India
6 Department of Chemical Engineering, National Institute of Technology, Srinagar 190006, Jammu and Kashmir, India
7 Independent Researcher, Sharjah 27272, United Arab Emirates
8 School of Earth and Environmental Sciences, Queensland University, St. Lucia, QLD 4067, Australia
* Correspondence: fayaz.env@jk.gov.in (F.A.M.); a.halog@uq.edu.au (A.H.)

**Abstract:** During the last half-century, the CO₂ concentration in the world's atmosphere has increased from 310 p.p.m. to over 380 p.p.m. This is due to the widespread usage of fossil fuels as a main source of energy. Modeling forecasts have shown that this trend will continue to rise and reducing CO₂ emissions is a challenging task for multi-stakeholders, including research institutions. The UN Climate Change Conference in Glasgow (COP26) has stressed that stakeholders need to work together to achieve a NetZero target. Technologies involving absorbents for the capture of CO₂ from a gas mixture are energy-intensive. Carbon adsorption and conversion (CAC) approaches have been gaining attention recently since these technologies can mitigate CO₂ emissions. In this review, materials ranging from advanced carbon-based materials to natural resources-based materials will be reviewed. Adsorption and conversion capacities as well as the scalability possibility of these technologies for solving the CO₂ emission problem will be investigated. The review, therefore, is timely and meaningful concerning the net zero emission targets set by countries and developmental organizations worldwide.

**Keywords:** carbon-based materials; CO₂ adsorption; CO₂ conversion; fossil fuel; climate change; global issues

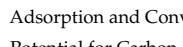



## 1. Introduction

Recently, anthropogenic global warming and climate change have become global issues. Carbon dioxide (CO₂) is a greenhouse element, and evidence indicates that human-caused CO₂ emissions are a major contributor to global warming [1]. It is forecasted that fuel-based power plants alone will continue to increase in number. Drastic changes in the global climate and a significant increase in global temperature have been reported. There were 315 natural catastrophes in 2018 worldwide, most of which were weather-related. About 68.5 million people were impacted by natural catastrophes such as cyclones, tornadoes, forest fires, and droughts, resulting in a total estimated cost of USD 131.7 billion in damages [2]. It is extremely worrisome that 2018's economic losses from fires are nearly equal to those suffered due to wildfires over the entire previous decade. Food, water, health, the environment, and infrastructure are some sectors most affected by climate change [2].

Fossil fuels are currently some main sources of artificial $CO_2$ emissions [3], and capturing $CO_2$ is becoming very important globally. In 1977, it was proposed that $CO_2$ could be removed from the exhaust of a coal power station and stored in a rock formation. This idea gave rise to the field of carbon capture and storage. According to the International Energy Agency, this method can cut emissions of $CO_2$ by 17 percent by the year 2050, so it should be incorporated into the policies of every country to lessen the impact of global warming. Adsorption, membrane separation technology, absorption, and cryogenic processes are common methods that can be used to store and treat $CO_2$ [4]. Studies on $CO_2$ adsorption and conversion have been completed and reported in the literature (details provided in Table 1 below). In addition, activated carbon, porous silica [5], zeolite [6], sepiolite [7], metal–organic frameworks (MOFs) [8], boron nitride (BN), Mxenes [9], ionic liquids, and porous adsorbents are some of the known materials for $CO_2$ adsorption [10]. However, a thorough review of carbon-based materials synthesized from bio-sources, their adsorption mechanism, preparation methods, their characteristics, and some industrial scale-up of these technologies are still lacking and require further investigation. This review aims to shed light on these points as well as to discuss state--of-the-art technologies in $CO_2$ capture and conversion fields.

**Table 1.** Reviews of bio-based synthesized materials.

| Topics Reviewed | Year Published | Ref. |
| --- | --- | --- |
| Seaweed functionality as a sustainable t | 2021 | [11] |
| Bio-based materials such as bio-waste, their modification technique, and their potential application as a sorbent material for energy | 2016 | [12] |
| Biomaterials, their unique properties, and examples of them that can potentially be used for $CO_2$ removal. | 2013 | [13] |
| Surface-modified activated carbons as sustainable bio-based materials for environmental remediation | 2021 | [14] |
| Bio-based carbon materials for anaerobic digestion | 2021 | [15] |

## 2. $CO_2$ Adsorption Mechanism

To investigate adsorption processes, it is important to learn about the adsorption mechanism of different materials. The quantum sieve impact, kinetic impact, dynamic impact, and molecular sieve impact are some of the adsorption techniques suggested [16]. When separating $H_2$ and $D_2$ using a quantum sieve, for example, the effect is strongly related to the de Broglie wavelength and the diffusion velocities of the gas molecules [17,18]. When all of the molecules in the gas can travel through the adsorbent pores, the polarizability, dipole moment, and magnetic sensitivity of the adsorbate and adsorbent surfaces play a much larger role in the capture of specific species [19,20]. This is because the dynamic effect dominates the adsorption behavior under these circumstances. The kinetic diameter (i.e., collision distance) is the minimum distance between the adsorbate and adsorbent molecules approaching each other with zero kinetic energy [21,22]. This distance determines the efficiency of $CO_2$ capture under the carefully regulated pore size of adsorbents (such as zeolites and carbons). Because of its molecular filter mechanism, this process is extremely temperature dependent. In air separation and $CO_2$ capture, separating $CO_2$ from $CH_4$ requires fine-tuning the adsorbents' particle size [23,24]. There are two main categories of contact between carbonaceous adsorbents and $CO_2$ molecules: physical and molecular. Typically, carbons physically adsorb $CO_2$ via the van der Waals force, which is closely linked with temperature and pressure [25] (i.e., the process by which pores are filled). High pressure and low temperature significantly improve the adsorption ability [26,27]. The $CO_2$ molecules are chemically bound to the adsorbent surface, making the connection greater than that of physical capture. For chemical adsorption, the formation of hydrogen bonds

and the interaction between acids and bases are two important processes. The graphical mechanism of $CO_2$ adsorption is illustrated in Figure 1.

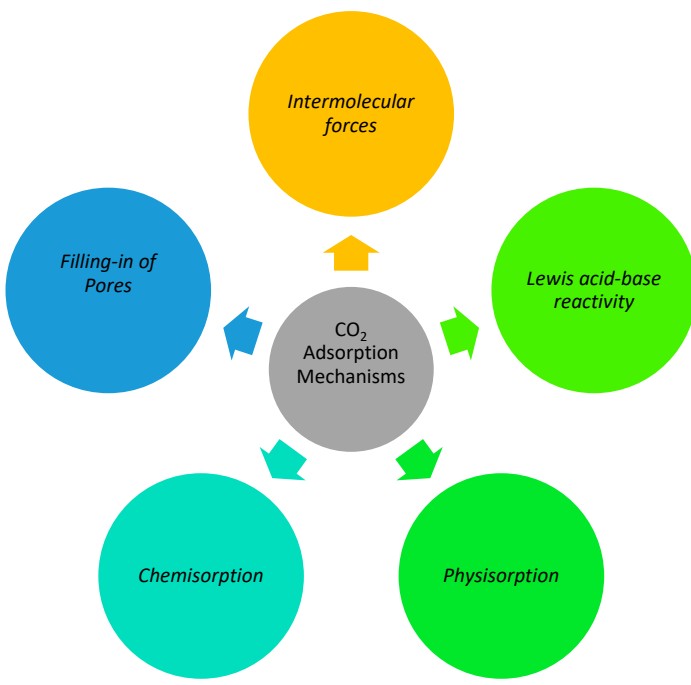

**Figure 1.** Mechanisms of $CO_2$ Adsorption.

### 2.1. Filling-In of Pores Mechanism

When the pore size of the adsorbent is about the same as the $CO_2$ molecule, more $CO_2$ will be absorbed [28] The kinetic diameter of $CO_2$ is predicted to be 0.33 nm, making it significantly lower than that of $CH_4$ (0.38 nm) and $N_2$ (0.5 nm) (0.364 nm). For optimal $CO_2$ selectivity, the adsorbent's pore size should be controlled to be between 0.33 and 0.36 nm. On the other hand, it was discovered that $CO_2$ adsorption increases in proportion to total micropore volume [29]. An increase in $CO_2$ adsorption was seen at a decreased pore volume, indicating that pore filling regulates $CO_2$ adsorption by ultra micropores. In the experiment by Parshetti et al. [25], the adsorption capacity for $CO_2$ increased with both the surface area and the micropore volume. This suggests a unique pore-filling process and physical adsorption behavior.

### 2.2. Intermolecular Forces

Hydrogen atoms form covalent links known as hydrogen bonds when they join with other elements that can accept hydrogen. The oxygen atoms in $CO_2$ are negatively charged because of their large electronegativity gap with the carbon atom, enabling them to make weak hydrogen bonds with CH on the carbon's surface [26]. The behavior of $CO_2$ adsorption on N-doped carbon surfaces was analyzed using Niwa's model [30]. The high electronegativity of oxygen in $CO_2$ makes the CHO hydrogen bond weaker than the OHO and NHO bonds. The redshift of the expanded CH anti-symmetric vibration point in the FT-IR spectra provided further evidence of hydrogen bonding. The surface hydroxyl group on the polybenzoxazine-based porous carbon used in another study was found to be responsible for the enhanced $CO_2$ adsorption capacity at reduced binder energy due to the presence of hydrogen bonding between the OH and the O atoms of $CO_2$ [31].

### 2.3. Lewis Acid-Base Reactivity

When forming a covalent coordinate connection, Lewis acids and bases interact with one another through the transfer of lone electron pairs from the acid to the base. For instance, carbon dioxide ($CO_2$) binds strongly to a nitrogen-rich carbon surface created

by the alkaline activation of an N-containing polymer. This is because the nitrogen-rich carbon surface's enhanced basicity and higher electron density bind the electron-deficient $CO_2$ molecules via a Lewis acid-base interaction [32]. Similar to N-doped carbon, an acid-pretreated biomass carbon adsorbent has several oxygenated groups on its surface. The adsorbent was able to bind additional $CO_2$ molecules because of the Lewis acid-base interaction since these groups took electrons from their neighbors to create a donor. The surface decorating of electronegative atoms created by in situ breakdown or pretreatment, as well as the addition of metal oxides, allows for the modification of the basicity and electron density of carbon [33]. Interactions between $CO_2$ and the low coordination number of $O_2$ ions found in metal oxides resulted in the formation of surface carbonates [34,35].

### 2.4. Physisorption

Physisorption, also known as physical adsorption, is the primary process accountable for $CO_2$ collection on activated carbon and char surfaces. Intermolecular contacts, such as the weak van der Waals force, are responsible for the adsorption of $CO_2$. These results are bolstered by the reality that adsorption can be undone. No initial energy is needed for the adsorption of $CO_2$ onto a solid char surface, and the physisorption of $CO_2$ molecules on the solid char surface is temperature dependent. At a lower temperature, more $CO_2$ is adsorbed at equilibrium). $CO_2$ has a significant quadrupole moment but no dipole. Since $CO_2$ has a straight structure with polar bonds at both ends, it can engage with active sites on the char surface. $CO_2$ adsorption via the physisorption adsorption mechanism is highly surface area dependent. It is important to note that the adsorption process also involves nitrogenous compounds such as amines, amides, and nitriles. Because of the powerful interactions between the acidic $CO_2$ and the basic nitrogenous surface functional groups, the presence of nitrogenous functional groups on char surfaces can improve the physisorption of $CO_2$ [36].

### 2.5. Chemisorption

An acidic $CO_2$ molecule forms a covalent connection with the active site of the adsorbent during chemisorption. Many different chemisorption mechanisms are seen in adsorbents. $CO_2$ adsorption, for example, is mediated by metal sites and surface functional groups in metal-organic frameworks (MOFs). The incorporation of nitrogen-containing functional groups into the char structure by the addition of amine-containing molecules can also lead to chemisorption [37]. This happens when water is added to $CO_2$, turning it into bicarbonate and producing a zwitterion (either carbamate or ammonium zwitterion). Biochar and hydrochar both have functional groups that may chemisorb $CO_2$, and these groups can be present in the biomass naturally or added by chemical modification, such as the addition of metal oxides [33,38].

## 3. Biomass-Derived Carbonaceous Materials

### 3.1. Various Bio-Based Synthesized Materials Have Been Developed

Carbon materials derived from biomass, agricultural by-products, and organic sources can selectively and efficiently capture $CO_2$, and the possibility that nitrogen is a major factor hindering selectivity is very interesting and meaningful to study. Chen Jin [39] studied the adsorption of $CO_2$ from sawdust waste at ambient conditions and increased the adsorption capacity by increasing N and microalgae (at a rate of microalgae per poplar sawdust = 0.5:1), and noted an adsorption result of 4.14 mmol $g^{-1}$ for $CO_2$ uptake, which is almost 1.7 times higher than non-N-enhanced adsorption (N-doping). Karolina Kiełbasaa et al. [40] studied the conversion of olive pulp into activated biocarbon for $CO_2$ adsorption by thermochemical methods, in which the author developed three types of porous activated bio carbons prepared for CO adsorption. Xuping Yang [41] studied the adsorption properties of biochar synthesized from seaweed-based biochar for $CO_2$ gas. Various carbon feedstock, pre-treatment methods, and $CO_2$ uptake are described in Table 2.

**Table 2.** Carbonaceous materials synthesized from biomass, organic sources, and adsorption capacity.

| Carbon Feedstock | Carbonization Conditions | Activation Method | $CO_2$ Uptake mmol $g^{-1}$ at 25 °C, 1 Bar | Ref. |
|---|---|---|---|---|
| Lotus leaves | 550 °C, $N_2$ atmosphere | KOH | 3.67 | [42] |
| Pineapple waste | 500 °C, $N_2$ atmosphere | $K_2C_2O_4$ | 2.22 | [42] |
| Potato starch | 800 °C, $N_2$ atmosphere | KOH | 2.80 | [42] |
| Cellulose | 800 °C, $N_2$ atmosphere | KOH | 2.80 | [42] |
| Sawdust | 800 °C, $N_2$ atmosphere | KOH and melamine | 2.20 | [34] |
| Palm date seeds | 900 °C, $N_2$ atmosphere | KOH | 4.36 | [42] |
| Guava seeds | 900 °C, $N_2$ atmosphere | KOH | 3.02 | [42] |
| Olive pomace | 600 °C | Steam and $K_2C_2O_4$ | 2.63 | [40] |
| Amazonian nutshells | 800 °C | KOH | 3.67 | [43] |
| Cotton stalk agro-residue | 700 °C | KOH | 4.24 | [44] |

### 3.2. Synthesized Carbonaceous Materials for $CO_2$ Adsorption

$CO_2$ levels have risen dramatically as a result of industrialization and economic development, contributing significantly to climate change and global warming. To counteract this, scientists have developed and begun using carbon-based adsorbents that are both effective and long-lasting in their ability to reduce $CO_2$ levels [45]. High adsorption capacity, long-term stability, high selectivity [45,46], high porous volume, and recycling ability at low chemical and energy usage costs are all desirable qualities in a $CO_2$ adsorbent substance. Carbonaceous materials are promising for carbon dioxide ($CO_2$) capture and storage due to their exceptional potential as adsorbents for $CO_2$ due to their morphological tunability, high specific surface area, functional groups, ample porous structure, exceptional gas storage capacity, reproducibility, and electronic properties [47,48].

To increase $CO_2$ capture efficiency, carbon-based materials can be modified in several ways, such as through material activation or N-doping. Activated carbon, carbon nanotubes, carbons derived from coal (e.g., bio-chars), carbons derived from metal-organic frameworks, carbon aerogels, carbons derived from polymers, and graphene oxide are just some of the carbonaceous materials that have been fabricated and used as $CO_2$ adsorbents [49].

Activated carbon (AC) is a porous substance that is widely used, and it can be produced from bio-waste. AC is a very potential for $CO_2$ capture, especially in humid environment due to its synthesis technique, low cost, high surface area, and hydrophobic characteristics [49,50]. To improve the structural properties of carbon materials and thus their $CO_2$ uptake performance, two steps are required in its production processes: (i) the carbonization of the precursor materials, which can be accomplished through the pyrolysis of biomass or hydrothermal treatment, and (ii) activation, which can be accomplished through physical or chemical methods [51]. Precursors such as pine nut shells [52], pine sawdust [43], water chestnut [53], the stone of the yellow mombin fruit, the stone of the acai fruit [51,53], Amazonian nutshells [43], and cotton stem agricultural residue [44] have been researched. Serafin et al. [54] used a one-step carbonization method supplemented with a chemical activation process using potassium hydroxide (KOH) to extract inexpensive carbons from walnut shells. The specific surface area increased up to 1868 cm$^2$/g, the micropore concentration increased to 0.94 cm$^3$/g, and $CO_2$ uptake capacity increased to 9.54, 5.17, and 4.33 mmol/g at temps of 0, 25, and 40 °C, respectively, in the resulting activated carbon (AC-800). Synthesized carbonaceous materials used for $CO_2$ adsorption are illustrated in Figure 2.

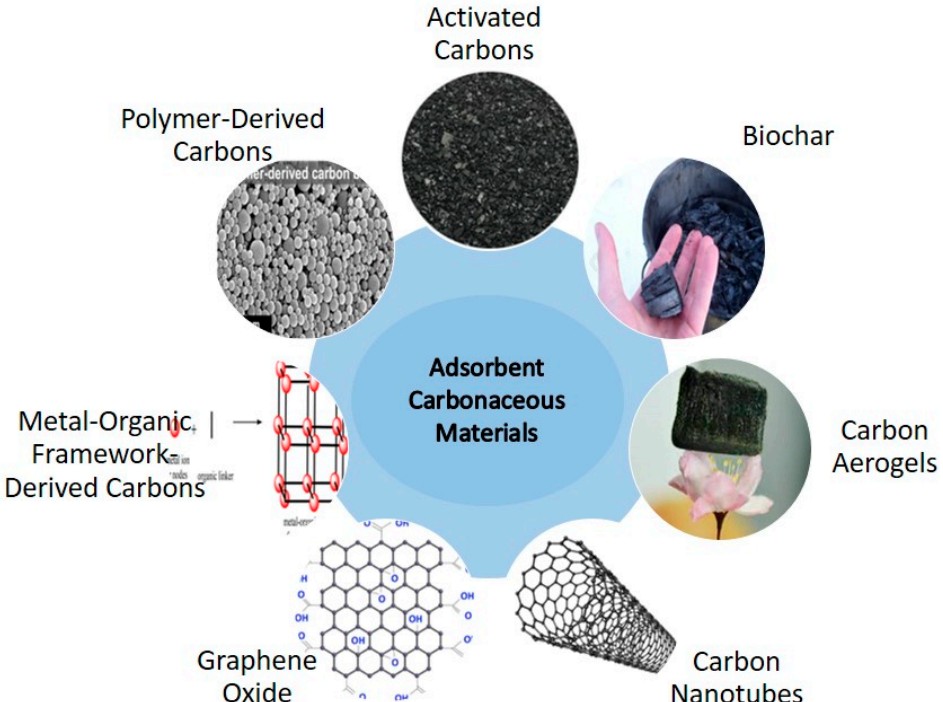

**Figure 2.** Synthesized carbonaceous materials used for $CO_2$ adsorption.

(a) Carbon nanotubes (CNTs) are hollow cylinders that have attracted attention as potential $CO_2$ adsorbents due to their exceptional water resistance, high mechanical strength, and efficacy in adsorbing $CO_2$ under high-pressure circumstances [55]. However, under low pressure, their adsorption capacity drops. The $CO_2$ adsorption efficacy of CNTs can be improved through chemical functionalization or composite adding. Including amine groups in CNTs resulted in more binding sites for $CO_2$. In the same way, the $CO_2$ adsorption capacity of CNTs can be improved by introducing other species, such as diamine or tetraethylenepentamine (TEPA) [55].

(b) Carbons produced from coal such as coal, coal tar, and pitch materials are inexpensive, readily available, and have a high tensile strength. These features make them promising materials for producing activated carbon (AC), as they have extremely large pores [56]. In addition to the efficient conversion of industrial byproducts, petroleum wastes have also been successfully changed into ACs using well-defined activation conditions. These ACs' hierarchical porous structures make them excellent $CO_2$ adsorbents over a broad pressure range. When used in a low-pressure range (1–35 bar) and at 25 °C, the activated adsorbents produced from anthracite coal significantly increase the $CO_2$ adsorption capacity [57]. The adsorption $CO_2$ uptake capacities are up to 10.51 mmol/g and 27.58 mmol/g [57]. This shows that ACs made from anthracite coal can improve their $CO_2$ adsorption capacity as a result of its high surface area.

(c) Carbons produced from metal-organic frameworks (MOFs): The use of MOFs in gas capture has received considerable attention due to their surface chemistry and extraordinary porosity [58]. A high concentration of metal sites within a MOF's framework significantly increases its binding capacity [59]. The existence of metal nodes and organic linking groups in a MOF allows the hole sizes and the 3D network structure to be tuned [60].

(d) Carbon aerogels (CAs) are porous materials that are perfect for gas adsorption because of their uniform pore distribution and tiny microporous structures. A study [61] showed that CAs' binding characteristics can be enhanced by the incorporation of heteroatoms and metals. For these reasons, CAs are a promising material for gas adsorption. The effectiveness of CAs depends on two factors: (i) the techniques used to prepare them, and (ii) the precursors chosen. By using inexpensive bio-resources,

Geng et al. [62] were able to create monolithic biocarbon aerogels with a meshed asymmetrical, hierarchical porosity structure. The bio-based carbon adsorbent was shown to have an adsorption capacity of 4.49 mmol g$^{-1}$ at 298 K and 100 kPa after the material's processing conditions were optimized [63]. Big mesopores (24.7 nm) and the distributed N moieties [64] formed after the surface modification of CAs (with N functionalities) with tetraethylenepentamine (TEPA) were found to increase the $CO_2$ adsorption by 4.1 mmol/g.

(e) Carbons produced from polymers have attracted a lot of interest because of their malleability in terms of structure, permeability, and functionalization [65]. Utilizing proper carbonization and activation techniques, a wide range of porous and heteroatom-doped carbon adsorbents can be synthesized from polymers. Using polymeric precursors containing nitrogen-based functional groups, a series of nitrogen-doped carbonaceous materials were synthesized. Post-polymer alteration [66] and monomer selection for high N content can also increase the nitrogen content of the final carbons [67].

(f) Graphene oxide (GO): GO has been proven to be a superior adsorbent. GO is created through an oxidation process, and it has a high porosity, a large surface area, and an excess of oxygen groups [68]. Basic groups introduced to the GO surface, on the other hand, can increase its $CO_2$ adsorption capability [69]. Polymers, such as polyetherimide, can have their binding capability increased by incorporating them with amine groups. Because of the N-rich surface created when polyetherimide (PEI) was impregnated onto the GO, its $CO_2$ adsorption capacity was enhanced via electron acceptor-donor interaction; this effect was further amplified when more PEI was added because more chemical linkages were created in the form of the carbamate complex [70]. Due to its powerful acid-base interaction, PEIGO was found to have a large uptake of $CO_2$ (84 mg/g). $CO_2$ adsorption enhancement through the creation of GO/metal heterostructures has been demonstrated [71]. Chen et al. [72] deposited Li and Al onto GO because of their lower toxicity, lower cost, and lower environmental impact compared to Ti. A higher $CO_2$ binding energy was achieved by anchoring the metals to the GO surface with urethane and hydroxyl groups.

Though those materials are promising concerning $CO_2$ adsorption in principle at research and lab scales, and some $CO_2$ adsorption setups have been used at the industrial scale, the specific $CO_2$ adsorption performance of those materials at industrial scales is t yet to be determined and needs to be tested and verified before being used on a larger scale.

## 4. Carbon-Based Materials for $CO_2$ Conversion

The global climate is changing unprecedentedly, and the primary reason behind this catastrophic change is the significant rise in global $CO_2$ emissions. It is critically important to lower the $CO_2$ concentration in the atmosphere, and this can be accomplished by $CO_2$ conversion technology where carbon-based materials are used. $CO_2$ conversion refers to the process of converting $CO_2$ into useful chemicals and fuels, and some common conversion methods are illustrated in Figure 3:

- Direct Air Capture (DAC): This is a potential technology that directly captures $CO_2$ from the air, and then converts it into useful chemicals and fuels. Industrial applications and the details of this technology are discussed in Section 5 of this paper.
- Biological conversion: This method uses microorganisms or enzymes to convert $CO_2$ into useful products (biofuels, food ingredients, and industrial chemicals) [73].
- Thermal conversion: This method uses heat to convert $CO_2$ into syngas (mainly $H_2$ and CO), methanol, and formic acid [74].
- Photocatalytic conversion: This method uses light energy and catalysts to convert $CO_2$ into various chemicals such as methanol, formic acid, and others [75].
- Electrochemical conversion: This method uses electricity and catalysts to convert $CO_2$ into various chemicals. This is the most widely studied and developed method that

converts $CO_2$ into value-added chemicals and fuels with the help of an electro-catalyst, often a metal or metal oxide [76].

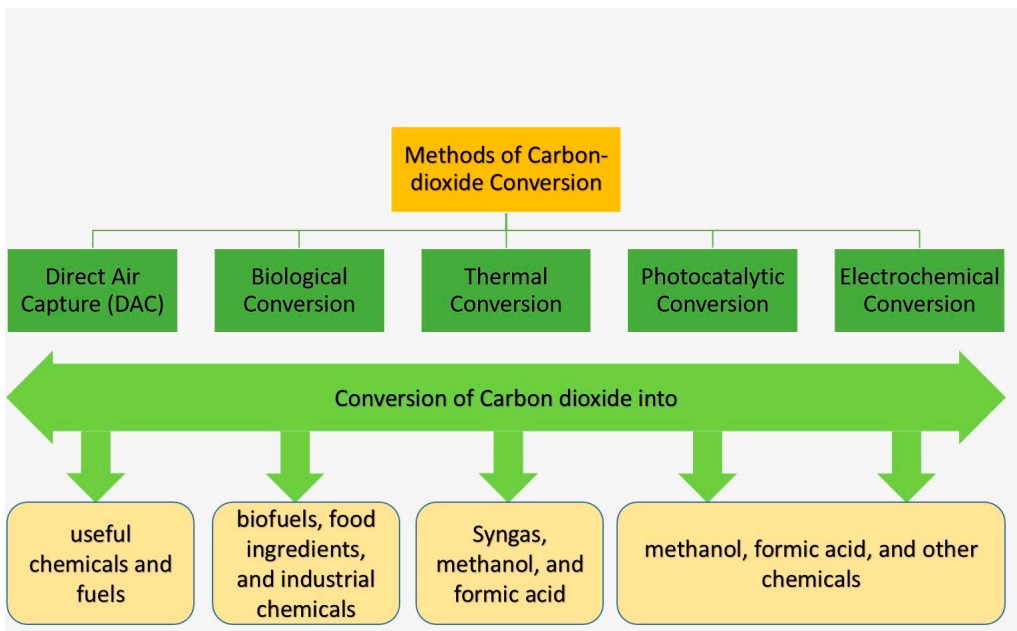

**Figure 3.** $CO_2$ conversion methods.

Different types of materials are potentially used for $CO_2$ conversions into useful chemicals and fuels, including carbon-based materials, namely activated carbon, graphene, and carbon nanotubes. These materials possess a high surface area and can act as catalysts, making them promising for use in $CO_2$ reduction reactions. However, more research is needed to optimize their performance and make the process more efficient and cost-effective.

*4.1. Graphene*

Graphene has been studied for its potential in converting $CO_2$ into useful chemicals and fuels. One way for $CO_2$ conversion using graphene is through graphene-based catalyst usage, which can facilitate chemical reactions that convert $CO_2$ into $CH_3OH$ or formic acid. Another application is membranes synthesized from graphene, which can selectively separate $CO_2$ from other gasses. Graphene has several qualities that make it a promising material for $CO_2$ conversion, and some of them are as follows [77]:

- High surface area: This characteristic makes graphene useful for catalytic reactions;
- High conductivity: Graphene is an excellent conductor of electricity and heat, which makes it useful in electrochemical reactions;
- Chemical stability: Graphene is chemically stable, and can be used in harsh environments and high-temperature reactions;
- Selectivity: Graphene membranes can be made to be highly selective, which makes graphene useful for $CO_2$ separation;
- Low cost: Graphene is made of carbon, which is abundant and inexpensive;
- Durability: Graphene is a strong and durable materials that can be used in long-term applications. These qualities make graphene a highly promising material for $CO_2$ conversion.

*4.2. Carbon Aerogels (CAs)*

These materials are highly porous, lightweight, have a high surface area, are usually made from carbon nanostructures, and have been shown to have the potential for $CO_2$ conversion [49,77]. Their high surface areas allow these materials to adsorb large amounts

of $CO_2$, and they, therefore, can be used in a variety of chemical reactions to convert $CO_2$ into other compounds [77].

One of the main ways ACs have been used for $CO_2$ conversion is in the form of adsorbents. $CO_2$ adsorption on carbon aerogels can be done through physisorption or chemisorption processes, which are then followed by the release of $CO_2$ by heating or by changing the pressure.

High surface area: CAs possess a very high surface area, typically from 300 to 2000 square meters per gram. This property allows them to adsorb large amounts of gases, liquids, and other substances [78];

- Low density: CAs are extremely lightweight, with densities as low as 0.003 g per cubic centimeter;
- High thermal conductivity: CAs have high thermal conductivity, which makes them useful for thermal insulation and heat dissipation;
- Mechanical strength: CAs have low compressive strength, but their mechanical properties can be improved by adding a binder or by using a different manufacturing process [79];
- High electrical conductivity: CAs can be made to be highly conductive, which makes them useful in applications such as super-capacitors and batteries;
- Porous structure: CAs have a highly porous structure and large pore volumes, which allows for the easy diffusion of gases and liquids;
- Low cost: CAs can be made from inexpensive, abundant materials, and their production process is relatively simple, which makes them a cost-effective material. However, these properties can vary depending on the specific type of carbon aerogel and how it was manufactured;
- Chemical stability: CAs are chemically stable and can withstand high temperatures and harsh environments [80].

### 4.3. Activated Carbons (ACs)

Activated carbons, usually known as activated charcoal, are carbon-based materials with a highly porous surface area which makes them useful for $CO_2$ conversion. ACs can also be used as a catalyst to convert $CO_2$ into other chemical compounds. For example, they have been used to catalyze the conversion of $CO_2$ into formic acid and methanol, which are useful chemicals for other industrial processes [81].

ACs can be used as a $CO_2$ adsorbent, which involves both the physical and chemical adsorption processes. The $CO_2$ can then be released by heating or by changing the pressure, making it possible to capture and store $CO_2$ in this way. ACs are particularly effective at adsorbing $CO_2$ at high pressures and temperatures, making them a potential material for capturing $CO_2$ from industrial processes [82].

Though those materials are promising in $CO_2$ conversion at research and lab scales, the $CO_2$ conversion performance of those materials at industrial scales still need to be tested and verified.

Yellow tuff, a natural tuff, and low-cost adsorption material, has been reported to be conveniently employed in a vacuum swing for $CO_2$ adsorption processes [83]. MOFs are proving to be effective adsorbents for $CO_2$ capture due to their microporous structure and chemical and thermal stabilities [84]. MOFs are capable of providing both physical and chemical interactions with $CO_2$ [83], while some other chemical adsorbents, such as amine-functionalized adsorbents, are capable of interacting strongly with the acidic $CO_2$ molecules [83]. Zeolites have been shown to have significant potential for $CO_2$ adsorption due to their high porosity, ultra-small pores, structural diversity, high stability, excellent recyclability, and chemical reactivity [35].

### 5. Carbon Capture Technologies for Climate Change Mitigation

Three main strategies exist for mitigating the consequences of climate change (illustrated in Figure 4). Traditional mitigation efforts aim to reduce carbon dioxide emissions

through the use of de-carbonization technologies and methods such as renewable energy, fuel switching [85], efficiency increases, nuclear power, and carbon capture, storage, and usage. The risks associated with these devices are generally low and have been around for some time [86–88]. The second possibility involves the use of novel tools and methods. It is possible to remove $CO_2$ from the atmosphere and store it using negative emissions devices [86]. Numerous negative emissions techniques are already in use, including bioenergy carbon capture and storage, bio-char, enhanced weathering, direct air carbon capture and storage, ocean fertilization, ocean alkalinity enhancement, soil carbon sequestration, afforestation/reforestation, wetland construction/restoration, and mineral carbonation/biomass use in construction [89–91]. The third and final choice involves controlling the amount of radiation coming from the sun and the planet. The goal of using geoengineering techniques that rely on radiative forcing is to either keep our world's average temperature stable or to make our planet slightly cooler. The main geoengineering techniques for radiative forcing are stratospheric aerosol injection, marine sky brightening, cirrus cloud thinning, space-based mirrors, surface-based brightness, and various radiation management approaches [92,93].

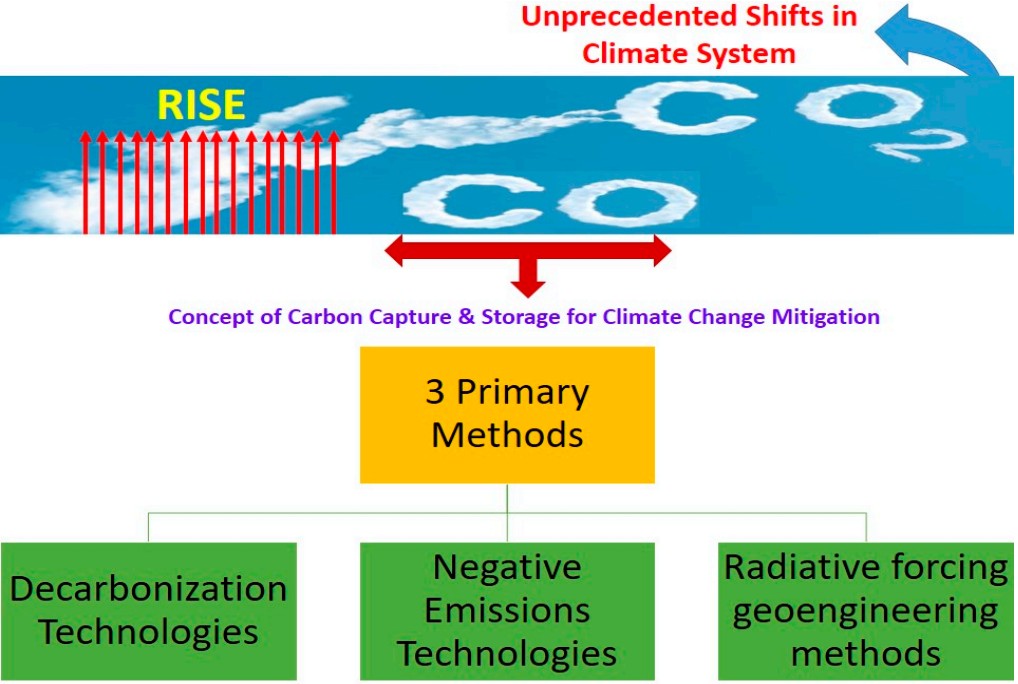

**Figure 4.** $CO_2$ conversion methods for climate change mitigation.

As a means of decarbonizing the energy and manufacturing industries, carbon capture and storage holds a great deal of potential. The burning of solid fuels such as coal, oil, or gas can produce carbon dioxide gas, which can now be collected and isolated using modern technology. The $CO_2$ is then transported through pipes to subterranean storage facilities, where it will stay for thousands of years. Eliminating or at least greatly lowering the pollution from burning fossil fuels is priority number one. The three possible capturing times are before, after, and during oxyfuel combustion. When it comes to $CO_2$ capture and storage, each method has its specific process. Post-combustion capture devices, on the other hand, are well-suited for retrofitted uses. $CO_2$ is captured, liquefied, and then transported via ship or conduit to long-term holding sites. It has been found that depleted oil and gas areas, coal beds, and underground saline reservoirs that are not used for potable water are all viable options.

*Direct Air Carbon Capture Technology*

One of the emerging technologies for artificially removing $CO_2$ from the atmosphere is direct air carbon capture and storage (DACCS). This technology uses molecular bonding to extract $CO_2$ from the atmosphere, and it is then either kept in geological reservoirs or put to some other use, such as in the production of compounds or mineral carbonates adsorption. The stability of $CO_2$ storage is a major issue for this technology as well, just as it is for carbon capture and storage and bioenergy carbon capture and storage. Capturing $CO_2$ from atmospheric air is much more challenging than capturing $CO_2$ from highly concentrated combustion gas streams since direct air carbon capture and storage requires more energy and materials. If unforeseen issues with large-scale implementation can be resolved, the global capacity for $CO_2$ removal will be between 0.5 and 5 $GtCO_2$ per year by 2050 and could increase to 40 $GtCO_2$ per year by the end of the century. The cost of $CO_2$ removal is estimated to run from USD 600 to 1000 per metric ton of moved ($tCO_2$) at the beginning but to drop to USD 100 to 300 per metric ton of $CO_2$ removed in the future [94]. Several companies have developed and implemented direct air capture technology at an industrial scale. These companies include Carbon Engineering, Climeworks, Global Thermostat, Hydro Cell, Sky Tree, and Infinitree. Carbon Engineering has constructed a DAC plant located in the Permian Basin in Texas, USA, with a capture capacity of 1 $MtonCO_2$/year [95]. There are no legislative instruments in place to aid this technology, as is the case with many of the other negative emissions options that have been investigated.

The direct air capture of $CO_2$ by physisorbent materials, namely TEPA-SBA-15 (amine-modified mesoporous silica, chemisorbent, and Zeolite 13X (inorganic), HKUST-1, Mg-MOF-74/Mg-dobdc, and SIFSIX-3-Ni (a hybrid ultra-microporous material including four physisorbents), were found to be able to capture carbon from $CO_2$-rich gas mixtures [96]. The greenhouse gas removal efficiency was 79–91%, while in the best case, the removal efficacy can be as high as 97% [97]. However, competition and reaction with atmospheric moisture significantly reduced the direct air capture (DAC) performance [96], and the drawback of DAC lies in the significant amount of energy required for capturing $CO_2$ from the atmosphere [98]. Amine-based cellulose adsorbents or silica gel are normally used in the DAC system [99,100]. An estimate of US$610–780/tC $CO_2$ was reported for the facility using sodium hydroxide, which is a strong base [95]. The thermal energy required for a DAC adsorption system is between 1400 and 2000 kWhth/ton $CO_2$ [100].

## 6. Carbon Recycling through $CO_2$ Conversion

Circular economy is a model that recycles all waste and byproducts discharged from a production process and generates no waste at the production end pipe. A cycle c economy is a process that involves the transforming of $CO_2$ from a linear economy into value-added chemicals and fuels. $CO_2$ is converted into methanol with enzymes or methane via electrodes and hydrogenophilic methanogenic cultures [101]. Catalysts can also be used in converting $CO_2$ into other chemicals. For example, activated carbons have been used as a catalyst in converting $CO_2$ into formic acid and methanol, as mentioned above, both of which are useful chemicals that can be used as feedstock for other industrial processes. About 200 million tons of $CO_2$ are utilized in different processes each year, and then $CO_2$ will react with ammonia to produce urea for fertilizers, while petroleum companies are also injecting $CO_2$ underground to help recover fossil oil. Methanol and $H_2$ are some of the key products produced from $CO_2$ conversion processes, and those chemicals can be green or gray methanol and $H_2$. When methanol is produced from renewable energy sources, a new term, "e-methanol", is used. In terms of environmental impact, $H_2$, and methanol are known as the lowest carbon intensity chemicals, and this is best for the climate. Methanol is one of the most practical alternatives to conventional fuels in the maritime area globally, and its use in a ship's operation can reduce $SO_x$ emissions by 99%, PM emissions by 95%, $NO_x$ by 60%, and $CO_2$ by 25% [102].

Recently, there has been an increasing number of projects that utilize sustainable feedstock such as captured $CO_2$ from industrial emitters and green hydrogen produced

from municipal solid waste (MSW), forestry residues, or agricultural waste. In terms of market impact, methanol is available in over 100 ports today. Conventionally, by 55% of global consumption of methanol is used in the production of downstream chemicals. Increasingly, the fastest growing segment is where it is in the numerous applications where it is consumed as a fuel (~45%). Captured $CO_2$ can also be used for enhanced oil recovery [103].

Using methanol as a fuel would produce $CO_2$ emissions of 54.7 tons per day at service speed, compared to 64.7 tons per day for diesel, and even less when using renewable or bio-methanol blends. Methanol is known to be a fuel that has no sulfur emissions, very low particulate matter, and $CO_2$ emissions 15% lower than conventional marine fuel oil. Methanol can even blend with water to meet IMO $NO_x$ Tier III requirements, removing the need for expensive exhaust gas treatment [104]. Renewable methanol can also be produced from renewable $H_2$ and captured $CO_2$. Globally, methanol was valued at USD 30.7 billion in 2021 and is forecasted to reach USD 36.3 billion by 2026. The option to recycle $CO_2$ could reduce industrial costs related to $CO_2$ certification, and in Germany alone, the production costs of e-MeOH or methanol produced from electricity will vary between EUR 608 and 1453 per ton.

Hydrogen can also be produced from captured $CO_2$. An electrolyte is used in this technology. When $CO_2$ is injected into the aqueous electrolyte, the $CO_2$ will react with the cathode, making the solution more acidic, which eventually generates electricity and creates $H_2$. This is a new trend and has a great environmental and market impact since captured $CO_2$ can be recycled and turned into a clean energy source [105].

## 7. Techno-Economic Analysis and Life Cycle Assessment

$CO_2$ capture is a potential approach, though its techno-economic feasibility and life cycle vary, depending on technology used. Some carbon-based synthesized adsorbents used in its $CO_2$ capture facility are expensive, while some adsorbents synthesized from bio-sources such as activated carbon seem to be less expensive due to the availability of bio-sources in various countries. Physical adsorbents such as zeolites are widely used in the oil and gas industry, and can adsorb as much as 6.18 mol $CO_2$/kg adsorbent at 25 °C and 1 bar [106]; however, these adsorbents have a low selectivity of $CO_2$ compared to other gases, they become negligible above 200 °C, and their adsorption capacity declines quickly with increasing temperatures above 30 °C [106]. For zeolites, the $CO_2$- capture efficiency could reach up to 96% [107]. Multi-walled CNT-synthesized adsorbents are cost-effective for $CO_2$ capture from flue gases, are stable for prolonged adsorption-desorption cyclic operation, can run for up to 20 cycles, and have a maximum $CO_2$ adsorption capacity of 2.61 mol $CO_2$/kg adsorbent at 20 °C and 1 bar [106]. ACs synthesized from the stones of yellow mombin are found to have 10 cycles [50].

Several $CO_2$ capture and conversion processes have been implemented, such as to produce synthetic natural gas [108], and this process is effective in reducing both the net $CO_2$ emission to the atmosphere by 45% and to have a cost of EUR 2.39 € per kg Raw-SNG [108]. The direct conversion of $CO_2$ into liquid fuels and natural synthetic gases [109] also shows that this technology is promising, efficient, and profitable. It was found that DAI units normally have a lifetime of 20 years [98]. A study by Sai Gokul Subravet [110] proved that the $CO_2$ avoided costs for a two single-stage pressure–vacuum swing adsorption (PVSA) system ranges from EUR 87.1 to 10.4 € per ton of $CO_2$ avoided; the PVSA system is attractive for high $CO_2$ concentration streams such as emission streams from steel and cement plants, while ideal adsorbents should have a wide range of $CO_2$ affinity and a negligible $N_2$ affinity [110]. A work by Anshuman Sinha on DAC [111] shows that the cost of DAC lies between USD 86 to 221 per $tCO_2$, while the thermal energy ranges between 3.4 to 4.8 GJ per $tCO_2$ captured, and the electrical energy ranges between 0.55 to 1.12 GJ per $tCO_2$ captured [111].

## 8. Challenges and Perspectives

$CO_2$ capture and $CO_2$ conversion have remained a trendy topic of discussion, research, and technology development of late due to the urgent need to address the problem of climate change. Achievements in the field have been reported. In terms of the conventional treatment of $CO_2$ in industry, polluted $CO_2$ is usually treated by the absorption method, which involves the use of a liquid phase solvent sprayed against the gas phase in a wet scrubber system. However, the technologies remain under development because of several challenges, some of which are as follows:

(a)  The mismatched scale of operations: $CO_2$ capture and conversion require a significant upfront investment, particularly for transmission. The storage infrastructure and conversion are also a challenge. The key to success (expressed in adsorption capacity or $CO_2$ separation efficiency) often depends very much on the adsorbent or membrane materials.

(b)  In the adsorption-based $CO_2$ capture processes, the choice of the gas-solid contact system (fixed bed, fluidized bed, moving bed) is crucial. Merely choosing the most promising adsorbent (i.e., the adsorbent providing the best combination of required properties) is not enough for the commercial deployment of the adsorption technology; the potential of each sorbent can be fully exploited only by using the most suitable combination of a gas–solid reactor configuration and regeneration mode [112–114].

(c)  The need for a social license to capture, transport, and store $CO_2$, in addition to ongoing storage liabilities, are some other challenges that need to be overcome to make these technologies fully usable in real applications. Capturing and converting $CO_2$ into other gasses remains a challenging task for many agencies, organizations, and stakeholders. There are several issues with carbon capture and storage, such as insufficiently protected storage capacity and the possibility of leakage. The environmental consequences of accidental leaks at coastal storage sites have been reported [115]. Additional concerns with this technology include public acceptability [94,116] and expensive execution costs. $CO_2$ capture enables a wide variety of applications, including chemical production, fuel generation, microalga cultivation, concrete production, and oil recovery [93,117].

$CO_2$ capture and conversion technologies have an important role in capturing and reducing $CO_2$ and combating climate change at the global level. $CO_2$ capture performed by biological approaches such as microalgae utilization to convert $CO_2$ into valuable products [118–123] and novel catalysts for improving the adsorption capacities of existing and bio-based materials might be new areas of research and development in this important field.

## 9. Conclusions

$CO_2$ is continuously released into the atmospheric environment via different routes, but mainly via industrial activities such as energy production, transportation, and oil and gas production. Increasing $CO_2$ emissions cause anthropogenic global warming. COP26 set a target for countries concerning their $CO_2$ emissions. $CO_2$ capture and conversion can be attainable by both conventional and advanced technologies, while adsorption and electrochemical methods have certain advantages. Synthesized carbonaceous materials such as silica-based adsorbents, polymeric materials, clay, biomass-based materials, carbon nanotubes and graphene oxide, and other materials have shown to be promising $CO_2$ adsorbents. Activated carbon can be prepared and activated by different methods such as physical or chemical activation processes, while carbon nanotubes can be produced through (i) arc discharge, (ii) laser ablation, and (iii) chemical vapor deposition. Some physical properties (surface area, porosity) and chemical properties (functional groups of the materials) are significant factors determining the performance of these carbonaceous materials. The availability of feedstocks such as agricultural wastes for producing these materials, and catalysts modifying and increasing adsorption capacities of these materials are prospective areas of research and development for $CO_2$ capture and conversion technology in the future.

**Author Contributions:** Conceptualization, T.-D.H. and S.A.B.; methodology, T.-D.H. and S.A.B.; validation, T.-D.H. and S.A.B.; formal analysis, T.-D.H., S.B. and F.A.M.; investigation, T.-D.H. and S.A.B.; resources, T.-D.H. and S.A.B.; data curation, T.-D.H. and S.B.; writing—original draft preparation, T.-D.H., S.B., F.A.M., I.Q. and S.B.P.; writing—review and editing, T.-D.H., S.A.B., F.A.M. and A.H.; visualization, T.-D.H. and S.A.B.; project administration, T.-D.H. and S.A.B. All authors have read and agreed to the published version of the manuscript.

**Funding:** This research received no external funding.

**Institutional Review Board Statement:** Not applicable.

**Informed Consent Statement:** Not applicable.

**Data Availability Statement:** Not applicable.

**Acknowledgments:** I would like to acknowledge the generous advisory support and opinion provided by Nhuan Nghiem of Clemson University, USA, and Le Minh Thang of the School of Chemistry and Life Science, Hanoi University of Science and Technology, who have encouraged me to structure and prepare this manuscript for submission. Sincere thanks also go to the RoHan Project funded by the German Academic Exchange Service (DAAD, No. 57315854) and to the Federal Ministry for Economic Cooperation and Development (BMZ) of Germany inside the framework of the "SDG Bilateral Graduate School program" for providing a. research scholarship to T.D.H so that he could focus on and complete this work.

**Conflicts of Interest:** The authors declare that they have no conflict of interest.

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
