# Peer review of "Carbon-Based Synthesized Materials for CO2 Adsorption and Conversion: Its Potential for Carbon Recycling"

_recycling, doi:10.3390/recycling8040053_

Round 1

Reviewer 1 Report

General comment:

The manuscript deals with an overview on adsorption and conversion capacity of carbon-based materials from natural resources for CO2 removal. Moreover, the possibility of scaling up these technologies was considered.

The manuscript is suitable to be published in this journal; however, some points should be addressed before publication.

Some minor language mistakes are present that should anyway be corrected.

1. Introduction

CO2 capture can be performed by several biological approaches, such as microalgae to convert CO2 into valuable products. I would like to suggest to consider the following manuscript:

·      Bench-scale cultivation of microalgae scenedesmus almeriensis for CO2 capture and lutein production (2019) Energies, 12 (14), art. no. 2806;

·      Biologically-mediated carbon capture and utilization by microalgae towards sustainable CO2 biofixation and biomass valorization – A review (2022) Chemical Engineering Journal, 427, art. no. 130884;

·      Advanced carbon sequestration by the hybrid system of photobioreactor and microbial fuel cell with novel photocatalytic porous framework (2021) Bioresource Technology, 333, art. no. 125182.

4. Synthesized carbonaceous materials for CO2 adsorption

This section is mainly qualitative. I would like to suggest to include operative conditions, CO2 removal performance and the scale (i.e. lab scale, pilot scale, industrial scale) for each material considered.

Please, clarify how the scale-up was considered.

5. Carbon-Based Materials for CO2 Conversion

This section is mainly qualitative. I would like to suggest to include operative conditions, CO2 removal performance and the scale (i.e. lab scale, pilot scale, industrial scale) for each material considered.

Please, clarify how the scale-up was considered.

Please, specify the chemicals/compounds that are produced from the conversion of the CO2 capture.

6. Carbon Capture in Climate Change Mitigation

Please, clarify how the scale-up was considered.

7. Carbon Recycling Through CO2-Conversion

This section is generic. Please, go in deep reporting the produced products, their impact on the market, the environmental aspects.

Author Response

Please kindly have a look at our attached file for our response to the 1 reviewer

Reviewer 2 Report

The work does not contribute any novelty in the field.

The introduction is misdirected, it does not fix any specific aspect that the authors wish to cover.

This inconvenience is general in the work, it covers too many aspects without delving into any of them.

Reviewer 3 Report

The authors reviewed on potential carbon-based materials, its adsorption and conversion capacity as well as the possibility of scaling up. This is an important topic, but there are several matters that needs to be addressed: 

1.      It is quite confusing that the terms of “adsorption” and “absorption” were interchangeably used e.g. in Figure 1. Please get the terms correct and specific for the meaning of the sentence because technically, these two are different.

2.      The transition to Section 2 is rather abrupt. Besides, whether it is chemisorption or physisorption depends on the type of adsorbent/absorbent used. It is not proper to generalize the situation as explained in the current manuscript is applicable for all.

3.      For Section 4 and Section 5: it is more on qualitative review. It needs to be improved with a clearer comparison on the effectiveness of each material.

4.      Section 7 needs to be expanded to balance up with other easier sections for coherence.

5.      No detailed insights can be obtained from this review paper.

       The manuscript needs to be proofread for proper punctuation.

Reviewer 4 Report

review attached as pdf

Round 2

Reviewer 2 Report

1.       The article has several spelling and punctuation mistakes.

2.       The introduction is misdirected, it does not fix any specific aspect that the authors wish to cover.

The Introduction should be focused on the different aspects that the work wants to cover. Mechanisms of adsorption, Design of carbonaceous materials ….The objective it is not clear.

3.       What is the difference between section  3. “Carbonaceous materials synthesized from biomass, organic sources” and section4: “Synthesized carbonaceous materials for CO2 adsorption”? Do you need two different headings?.

Isn't one heading "CO2 sequestration on carbonaceous materials" enough? This section should include a large number of references because the scientific papers on this subject are innumerable. Could you focus on what's new: Hybrid compounds, thermal and physical activation and link with the adsorption mechanisms described in the previous section.

4. Section 6. “Carbon Capture in Climate Change”. The reported data is not of interest in the middle of the article. Perhaps they could have been included in the introduction. In section 4, a proposal is expected, a scientific solution, even if partial, not a CO2 damage report. It makes you lose the objective of the work, which was already misfocused since the introduction. Check the title: Solutions should be focused on adsorption on carbonaceous materials.

5. Direct air carbon capture is an novel challenge that should be explained in detail. Effectiveness, risks, number of cycles in which the adsorbent can be used, actual efficacy and applicability data. It is not necessary to talk about types of adsorption again.

Reviewer 3 Report

a. The revised version is rather raw, e.g.

1. Section title and content are not properly segregated.

2. CO2 is not properly subscript.

3. There was a part with "....etc.,"

4. Figure 4 was not mentioned in paragraph.

b. DAC was depicted in both Figure 3 and Figure 4,  and it was later specifically explained under section for Figure 4. It should be integrated for a clearer picture of the position of DAC, rather than being segregated.

c. The title is about carbon-based material. But section 5 explained on the capability of MOF. The content doesn't seem to be aligned to the title.

d. The techno-economic analysis and life cycle assessment section seems to be rather surface. A more detailed insights should be provided to add value to this paper.

Reviewer 4 Report

The manuscript has been revised according to the reviewer’s comments. Therefore, it can be now accept for publication. 
